# Prototropy, Intramolecular Interactions, Electron Delocalization, and Physicochemical Properties of 1,8-dihydroxy-9-anthrone—DFT-D3 Study of Substituent Effects

**DOI:** 10.3390/molecules28010344

**Published:** 2023-01-01

**Authors:** Małgorzata Szymańska, Irena Majerz

**Affiliations:** Faculty of Pharmacy, Wroclaw Medical University, Borowska 211a, 50-556 Wroclaw, Poland

**Keywords:** 1,8-dihydroxy-9-anthrones, keto-enol equilibrium, QTAIM, aromaticity

## Abstract

1,8-dihydroxy-9-anthrone are tricyclic compounds with a ketone group in the middle ring and two hydroxyl groups substituted in the side-aromatic rings what results in formation of two intramolecular hydrogen bonds in which the oxygen atom from the ketone group is the proton acceptor. 1,8-dihydroxy-9-anthrones in which intramolecular proton transfer between C10 and CO in the middle ring occurs, can exist in a tautomeric keto-enol equilibrium. For anthralin, the most important representative of this group, this equilibrium has been studied previously, but it has not been studied for its derivatives. Substituents in the middle ring change the geometry of 1,8-dihydroxy-9-anthrones so they are also expected to affect the keto-enol equilibrium. It is also important to study the effect of intramolecular hydrogen bonds on the structure of both tautomeric forms. It was found that the nature of the substituent in the middle ring could affect the antioxidant properties of the investigated compound.

## 1. Introduction

In the previous work [1], we studied the influence of substituent on the structure of anthrones and anthraquinones-tricyclic compounds with a wide importance in biological processes. The change of the angle between the anthrone aromatic rings is associated with the change in electron density at the ring critical point of the central ring. 

An important group of anthrone derivatives are 1,8-dihydroxy-9-anthrones with substituents in the 1 and 8 position (Figure 1). Anthralin, the popular drug from the 1,8-dihydroxy-9-anthrones group, has two hydroxyl groups at position 1 and 8 on either side of the ketone group located at position 9 (Figure 1) and is used primarily in the treatment of psoriasis [2]. In the ketone form, it is stabilized by hydrogen bonds which are formed between the OH groups and the ketone oxygen [3]. For this reason, according to Hellier et al. [4], anthraline exists in its entirety in the ketone form [4]. The release of the hydrogen atom from the C10 position initiates the formation of free radicals, and although it may have an effect on skin irritation, it can be important in the mechanism of action of the drug [3,5]. Anthralin reduction power can be directly modified by changes in the structure [6]. For this reason, scientists are looking for new analogues substituted at the C10 position, which will prove to be more effective and, additionally, will not have side effects [5,7]. 

Intramolecular interactions of polycyclic compounds affect their structure [8,9,10]. The structure of a compound is related to the pharmacological activity; therefore, it is important to understand the effects of substituents, intramolecular hydrogen bonds, and keto-enol equilibrium on the structures of 1,8-dihydroxy-9-anthrone derivatives.

Two side rings of the 1,8-dihydroxy-9-anthrones derivatives are aromatic. The central ring in the lowest-energy ketone structure contains one CH_2_ group, which influences the aliphatic character of the ring. However, the remaining fragment of the central ring further delocalized as a result of π-π cross conjugation [11] and affects the cross delocalization of the other rings. The sensitivity of the middle ring to substitution can shift the character of the middle ring toward aromaticity, which can be affected by its non-planarity [1]. The presence of a ketone group in the middle ring and two adjacent OH groups in the side aromatic rings at the 1 and 8 positions causes the hydrogen atoms to form intramolecular hydrogen bonds. Migration of the proton between these ketone and hydroxyl group results in formation of tautomers [12,13,14,15]. Compounds, in which the intramolecular proton transfer takes place, are important in medicine and pharmacy and designing new drugs [16,17,18,19]. 

Keto-enol equilibrium can exist in 1,8-dihydroxy-9-anthrones through to a single proton transfer from C10 to CO in the middle ring (Figure 2). According to Marrero-Carballo et al. [15], for proton transfer to occur the central ring must be twisted into a boat-like conformation. This equilibrium is influenced by the solvents in which the substance is dissolved [13,20,21], the substituents [21,22,23,24], and the temperature [21]. Baba and Takemura [13] studied the keto-enol equilibrium between anthrone and anthranol-1 in isooctane using the spectrophotometric method. After dropping the anthrone in isooctane, no changes in the spectrum were observed even after two days. The reaction was much faster in the presence of a small amount of triethylamine, which is basic in nature [13]. Using computational methods, Marrero-Carballo et al. [15] compared the activation energy of intramolecular proton transfer in the chrysophanol molecule with the proton transfer to the pyridine molecule. They found that, in the case of intramolecular proton transfer in the anthrone molecule, the activation energy is much greater than in the case of proton transfer to the pyridine molecule. The higher activation energy is related to the deformation of the central ring in the anthrone molecule [15]. 

Laurella et al. [21] studied the effects of substituents on the keto-enol balance of β-ketoamides. They found that electron-withdrawing properties of chlorine atom causes an equilibrium shift towards the enol form, while methoxy groups that donate the electrons increases the content of the keto form. Additionally, they noticed that the equilibrium was also influenced by the position of the substituent, which is related to intramolecular hydrogen bonds in the molecule [21].

It is important to study the antioxidant properties of the analyzed compounds. Antioxidants have the ability to neutralize the harmful effects of free radicals. One of the methods of predicting the antioxidant properties is the determination of the enthalpy of the OH bond dissociation (BDE) and so the nature of the substituents in the ring affects the BDE value [25]. In the enol form, dissociation of the O-H bond at C9 is probable, which may have a positive effect on antioxidant properties. According to Lucarini et al. [26], electron-positive substituents decrease the BDE of O-H, while electron withdrawing substituents increase the BDE of O-H value relatively to unsubstituted phenol [26].

This work is a continuation of our previous [1] work in which we studied the influence of substituents on the structure of anthrones and anthraquinones and an analysis of 50 optimized compounds was performed. The substituents used in that work were differed in size, electron donating and electron accepting properties: NO_2_, CHO, COOH, CH_3_, CH_2_CH_3_, NH_2_, OH, Cl, C(CH_3_)_3_. In this work, each structure with a specific substituent has been optimized in the ketone form and in the enol form and the energy difference between them has been calculated. The lowest energy has been obtained for the keto structure with two hydrogen bonds and has been used as the reference energy. The electron density of the middle ring is sensitive to substitution as mentioned in the previous work [1,27] so also in this work it is used as a measure of substituent influence and aromaticity of the central ring. The process of intramolecular proton transfer is responsible for acid-base regulation in the cell or for enzymatic reactions, so it is important to study the effect of substituents on this reaction [12,15]. Moreover, the intramolecular hydrogen bonds have an influence on the geometry investigated compounds. If the substituents and hydrogen bonds affect the structure, then it is important to study their effect on the keto-enol equilibrium, which is the aim of the research undertaken.

### Computational Details

Optimization of the 1,8-dihydroxy-9-anthrone derivatives was performed with a Gaussian 16 package [28] at DFT-D3 B3LYP/6-311++G** level [29,30]. Grimme dispersion [31] was included to reproduce correctly the hydrogen bond in the investigated molecules. Vibrational frequencies were calculated to confirm that the optimized molecule reached the minimum of energy. QTAIM parameters were calculated with the AIMALL program [32] using the wave function for the optimized molecule.

## 2. Results

### 2.1. 1,8-Dihydroxy-9-anthrone Tautomers

To investigate the tautomeric preference, the Gibbs energy of different isomers in ketone form for four different substituents NH_2_, NO_2_, OH, and H has been compared (Table 1). There are six possible different tautomeric structures of the ketone form (Figure 3). For NH_2_, NO_2_, H, and OH substituted structures, five isomers with different energies have been obtained. In all cases, the K3 isomer converges to the K6 isomer, which is confirmed by the C9-O (1.261 Å), C9-C9a (1.462 Å), 4a-9a (1.412 Å), and 8a-10a (1.412 Å) bond lengths equal and respectively identical as these bond length for the K6 tautomer. The lowest value of the Gibbs energy for the K6 isomer has been obtained. Isomers with relative Gibbs energies higher than 10 kcal/mol can be neglected in the isomeric mixture owing to their exceptionally low percentage contents (<5 ppm). The smallest differences in Gibbs energy of 10 kcal/mol is obtained between isomers of NH_2_ and OH substituted structures. In all cases, the α angle is the greatest for the lowest energy structures. The HOMA values in the side rings of the preferred isomeric forms are always close to 1.

### 2.2. Crystal Structures

It can be expected that the tautomerism of dihydroxyantrone will be reflected in the structures of compounds deposited in the CSD database [33]. Since the structure of the model compounds is known [34], the analysis of the bond lengths in the 1,8-dihydroxy-9-anthrone derivatives in the crystalline state should allow to indicate which tautomer is realized in the crystal of individual derivatives. Appendix A summarizes the C-C bond lengths for the 1,8-dihydroxy-9-anthrone derivatives present in the CSD database. The table also includes CO bond lengths, HOC angles, and torsion angles allowing for the indication of proton deviation from the plane defined by the system of intramolecular hydrogen bonds typical of 1,8-dihydroxy-9-anthrones. The geometrical parameters should indicate which tautomeric forms are realized in the crystalline state.

The analysis of the CO bond system allows for a preliminary indication of the tautomeric form present in the crystal structures. In the structures K1, K2, and K3, the double bond of CO occurs in the side ring. The comparison of CO bonds in Appendix A clearly shows that all solid state structures can be of the K4, K5, or K6 type, in which the CO double bond connects oxygen to the central ring. According to the model structures (Figure 3), individual tautomeric forms should differ from each other in the arrangement of single and double bonds. The analysis of the CC bond lengths in the 1,8-dihydroxy-9-anthrone rings allows for the unequivocal elimination of the K5 tautomer, because all C3-C4 bonds are longer than those typical for benzene and are typical single bond. Since all the C2-C3 lengths in the analyzed compounds are shorter than the length typical for benzene (1.399) and the C1-C9a bond length is also shortened for a number of compounds, it can be considered that this structure is typical for solid-state 1,8-dihydroxy-9-anthrones. The analysis of C4a-C9a and C8a-C10a bond lengths indicates their shortening below the value typical for benzene in some dihydroxyantrone derivatives, which proves the possibility of the presence of the K6 tautomeric structure, characterized by the lowest energy. The linkage lengths of the dihydroxyanntron derivatives therefore indicate the possible presence of the K6 tautomer, with the K4 tautomer being more likely. However, the precision in determining the bond length should be taken into account. The comparison of the bond lengths in the central ring clearly shows that the K6 isomer dominates in the 1,8-dihydroxy-9-anthrone crystals. The shortest bonds, similar in length to benzene, are the bonds 4a-9a and 8a-10a.

It is a noteworthy fact that, in some structures, the protons of the hydroxyl groups remain unconnected with the carbonyl oxygen of the middle ring (BOLPEX [35], CARMYC11 [36], DHANQU03 [37], DHANQU04 [37], DHANQU08 [37], JUKREM [38], PIRFIH [39], QEGXUV [40], VURHEV [41]), as confirmed by the values of the torsion angles in Appendix A. The deviation of the proton from the plane convenient for the formation of intramolecular hydrogen bonds is due to the participation of the oxygen atoms in other interactions resulting from intermolecular interactions in the crystal.

### 2.3. Geometry of the Investigated Compounds

The α angles and relative energies for the optimized structures with different substituents are collected in Table 2. The lowest energy structure in the ketone form has been obtained for the K6 tautomer and remaining discussions in the study apply only to it. The Ea structure represents the enol form with one hydrogen bond and Eb with two hydrogen bonds. The Ka structure represents ketone form without hydrogen bonds, Kb with one hydrogen bond, and Kc with two hydrogen bonds. As in the previous works [1,27], the angle between the planes of the two side aromatic rings has been used as a measure of the substituent influence on the geometry of the ring system (Figure 4. Depending on the character of the substituent and its size as well as the presence of hydrogen bonds, the α angle changes from 0 to 52°. The H···O distance in the hydrogen bonds is close to 1.7 Å. Molecules in the ketone form, without hydrogen bonds, are characterized by the highest α angle. The angle decreases with the increasing number of the hydrogen bonds in the molecule. Deviations for this rule were observed for the structures of 9K6c, 9K6b and 5K6b, 5K6a. This is due to the different orientation of the substituent in the molecule. The α angle in the enol form is smaller than in the ketone form. The substituent always causes an increase of the α angle in the keto and enol form relatively to the unsubstituted molecule. The highest values of the α angle in keto form has been obtained for structure 4K6a and in enol form for the structure 4Ea. Both structures have a tertbutyl substituent so the increase of the α angle is connected with steric interaction of a bulky substituent. 

Table 2 shows the energy differences between the keto and enol form. For each substituent, the lowest energy has been obtained for the structure in the keto form with two hydrogen bonds.

In order to better understand the influence of substituents and hydrogen bonds on the aromaticity of the ring, the HOMA index was determined [42]. For the benzene aromatic ring, the HOMA index is equal to 1; HOMA=0 for the hypothetical structure of 1,3,5-cyclohexatriene with the reference C-C and C=C bonds of buta-1,3-diene [42]; and for the antiaromatic ring, it is negative.
(1)$$HOMA=1-\frac{\alpha }{n}\sum {\left(R_{o}-R_{i}\right)}^{2}$$
*R_o_*—the optimized CC bond length of a perfectly aromatic system equal to 1.388 Å.*R_i_*—determined bond length.*α*—standardization constant of 257.7.*n*—number of bonds.

The determined HOMA parameters for every ring of the investigated molecules are presented in Table 2. Additionally, as it was done in the previous work [1], to show the dependence of the particular geometry on the electron density, the value of the HOMA parameter of the middle ring has been correlated with the α angle. In the previous work [1], we determined the HOMA parameter for anthrones and anthraquinones with the same substituents in the middle ring. All compounds, irrespective of the nature of the substituent in the middle ring, which have OH groups in the side rings which do not form hydrogen bonds with the adjacent carbonyl group (enones), are characterized by the lowest value of the HOMA parameter for the middle ring. For the compounds investigated in this work the HOMA values for the middle ring are lower, if the OH groups in the aromatic side rings are not present. The presence of one or two hydrogen bonds in ketone form shifts the HOMA value of the middle ring towards higher aromaticity. The highest values of HOMA have been observed for all structures in the enol form with two hydrogen bonds. The electron density of the middle ring is related to the α angle and both parameters can be correlated (Figure 1). In Figure 1, the E and K6 structures have been used according to Figure 3. As the HOMA shifts towards higher aromaticity, the α angle decreases and hence the molecule flattens out. Similar conclusions were obtained in the previous work [1]. The highest values of the alpha angle for the enol form were obtained for structures with a bulky tert-butyl substituent. 

According to Ośmiałowski et al. [34], aromaticity and high π-electron delocalization in the whole system in the enol form plays an important role and determines the tautomeric preference of monohydroxyarenes. Nevertheless, high electron delocalization and stability in individual rings of the condensed systems also affect tautomeric equilibrium in the gas phase. The high HOMA in the side rings of the compounds affects the shift of the tautomeric equilibrium toward the ketone form.

### 2.4. Electron Density and Ellipticity at Bond Critical Point

The ellipticity of the electron density at the bond critical point (BCP) gives information about the nature of the C-C bond. A correlation between ellipticity and bond length for C8a-C9 and C9a-C9 has been found (Figure 2). The ellipticity and the length of the C-C bond in the middle ring is influenced by the form of the molecule is in and the number of the hydrogen bonds. No effect of the substituent on ellipticity and bond length has been noted. The highest ellipticity is observed for bonds in enol form. Suitable bond length of the C8a-C9 bond is in the range of 1.406–1.419 Å. The length of the C9a-C9 bond is in a slightly larger range of 1.405–1.424 Å. The smallest ellipticity and the highest C8a-C9 and C9a-C9 bond lengths are characteristic for the structures in the ketone form without intramolecular hydrogen bonds. The bond lengths of C8a-C9 and C9a-C9 have also been correlated with electron density at the BCP and the obtained correlation equations are: y = −2.1207x + 2.0518, R² = 0.9986; y = −2.2251x + 2.0799, R^2^ = 0.9971.

### 2.5. Selected Bands in the Theoretical IR Spectra of Dihydroanthrones

To study the effect of the substituent on the IR spectra, vibrational frequencies have been calculated for the favored structure in ketone form. In Table 3, the bands characteristic for C=O and OH groups are collected. For the NO_2_ substituent, the strongest νasOH band is observed at 3399 cm^−1^. This band repeats at a similar wavenumber for other substituents. Similar wavenumber repetition is observed for the vibration of νCO and δOH and the bands are located close to 1636 and 1670 cm^−1^. For the structure with the NH_2_ substituent, the vibration of νCO; δOH; δCH (C3, C4, C5, C6, C10) are observed at 1333 cm^−1^, which is not observed for other substituents.

### 2.6. Dipole Moment, Average Local Ionization Energy, and Electrostatic Potential

The dipole moment of the molecule provides important information about its structure. Table 4 shows the dipole moments of enol and ketone form with two hydrogen bonds. The dipole moments are in the range of 0.78–2.84 D. The highest value is obtained for the 9Eb structure. The COOH group significantly increases the polarity of the molecules. However, with the increase of the non-polar carbon chain, the polarity of the molecules decreases which can be observed in the structure with a tertbutyl substituted. Only in some cases (-Cl, CHO, NO_2_ substituents), the effect of the form of the compound on the dipole moment is observed. For the compounds with other substituents the values of the dipole moment in keto and enol form are very similar.

Average local ionization energy (ALIE) is a function to reveal regions containing highly reactive electrons in chemical system. The surface used for presentation of the local ionization energy is plotted on ρ = 0.0005 a.u. isosurface. Light blue spheres correspond to ALIE minima on the isosurface, revealing favorable sites of electrophilic attack. Darker blue color reveals relatively low ALIE regions. In this region, the electrons have relatively high reactivity.

In the structure of the hydrogen-substituted enol form, electrophilic attack is possible on C10 and on carbon atoms in the side rings, which are not common with the middle ring. In the enol form, the C10 atom is not subject to electrophilic attack. The electron donor substituent in the 2Eb structure directs the electrophilic attack to the side ring atoms that are not in common with the middle ring. In the ketone form 2K6c, the electrophilic attack is directed at the NH_2_ substituent. The OH substituent in the 3Eb structure revealed a favorable site of electrophilic attack on the carbon atoms of the side rings, which are not shared with the middle ring. In the enol structure, the site of electrophilic attack is the carbon atoms in the side rings that are not shared with the central ring. The site of electrophilic attack becomes less sensitive in the keto structure.

Electrostatic potential (ESP) is popular to visualize the electrostatic nature of molecules. It gives information about the chemical reactivity of a molecule indicating positively and negatively charged fragments of the molecule. For the structures in Table 4, the blue surface corresponds to minimal and red to maximal value of ESP. 

For the enol form, the minimum of ESP is located at the oxygen atom not accepting the proton what is connected with the oxygen lone pairs. In the ketone forms, the minimum of ESP is distributed on the oxygen atoms in the side rings OH groups at the and at CO in the middle ring. In addition, the minimum of ESP is observed on COOH, NO_2_, and Cl substituents and on the oxygen from CHO group. In the enol form, the maximum of ESP is accumulated on one of the hydrogen atoms in the OH group at the side ring.

### 2.7. Prototropy

Structures with OH and CHO substituents have labile protons, as they move around the molecule, can form a combination of different types of prototropic transformations [43]. Figure 3 shows all possible prototropic forms for OH-substituted 1,8-dihydroxy-9-anthrone. Five prototropic forms are obtained for the OH substituent and one for the CHO substituent (Figure 4). ΔE values, which are calculated relative to the isomer with the lowest energy, are placed under the structures. The highest energy for the OH substituent is obtained for the 3K5-C9a4aH structure, with a single bond between the C9a-C4a atoms. The most close in energy to the 3K6c structure is 3K6-C23H, which has a single bond between C2-C3 atoms. The only one prototropic transformation that has been obtained for the CHO substituent is very close energetically to the 8K6c structure (ΔE = 2.2 kcal/mol).

### 2.8. Antioxidant Activities

An important parameter for predicting antioxidant properties is the bond dissociation enthalpy (BDE) of the OH bond at C9 in enol form. The BDE of OH has been determined using the calculated total enthalpies (2). Additionally, BDE for C-H at C10 in keto form has been determined (3).
(2)$$BDE\left(O-H\right)=H(enolO^{\cdot })-H\left(H^{\cdot }\right)+H\left(enolOH\right)$$
(3)$$∆H=BDE\left(C-H\right)-BDE\left(O-H\right)$$

#### Investigation of the Substituents Effect on BDE (O-H) and BDE (C-H)

Table 5 shows the thermodynamic data for the tautomerization reaction. BDE values are given for C-H at C10 in keto form and O-H bond at C9 in the enol form. Structures with two and one intramolecular hydrogen bond have been considered. The low BDE(O-H) ensures easier detachment of hydrogen from the hydroxyl group. A low BDE(C-H) causes that the hydrogen attached to the C10 is easily removed by radicals [12]. A strongly electron-withdrawing substituent in the central ring like NO_2_ causes a significant increase of BDE(O-H) and BDE(C-H) in a structure with two intramolecular hydrogen bonds comparing to the unsubstituted structure. In structure with NO_2_ and one hydrogen bond slight increase of BDE(O-H) has been observed as well as a slight decrease of BDE(C-H) comparing to the unsubstituted structure. This means that the presence of the NO_2_ substituent and two intramolecular hydrogen bonds makes more difficult detaching the hydrogen atom from the OH and CH group. The electron donating substituent (NH_2_) causes a slight decrease in the BDE(O-H) value relatively to the unsubstituted molecule in structures with two and one intramolecular hydrogen bond. Thus, it is easier to detach the hydrogen from the OH group at C9 in enol form relatively to the unsubstituted structure. However, according to Korth and Mulder, a low BDE(O-H) alone does not make a compound a good antioxidant [12]. The opposite situation has been observed for BDE(C-H). The NH_2_ group caused a slight increase in the BDE(C-H) which means that it is more difficult to detach the hydrogen atom relatively to the unsubstituted structure. Similar BDE(O-H) results were obtained by Lucarini and Pedulli [26].

### 2.9. Hydrogen Bonds

Due to the presence of intramolecular hydrogen bonds, it is important to study their influence on the antioxidant potential and the tautomeric balance. In the conducted analysis, it is noticed that the BDE(O-H) and BDE(C-H) values are lower for the structures with two hydrogen bonds than with one hydrogen bond. Therefore, the presence of two hydrogen bonds increases the antioxidant properties of the compounds.

Hydrogen bonds have a significant impact on the enthalpy difference between ketone and enol structure. The free enthalpy difference between the ketone and enol form is greater for structures with two hydrogen bonds than for structures with one hydrogen bond. This means that two intramolecular hydrogen bonds have a greater effect on shifting the equilibrium towards the ketone form than the presence of one hydrogen bond.

To better illustrate the differences in energy associated with the presence of hydrogen bonds, Figure 5 shows a cycle of changes in energy for 1,8-dihydroxy-9-anthrone. When, in ketone form, the hydrogens of the OH groups in the side aromatic rings are directed towards the center ring, hydrogen bonds are formed. The energy differences between 1K6a⇆1K6b, 1K6b⇆1K6c, and 1Ea⇆1Eb are the hydrogen bond energies. Higher intramolecular hydrogen bond energy is observed for structures in the ketone form with two intramolecular hydrogen bonds. 

The energy of intramolecular hydrogen bonds is influenced by the substituents in the middle ring. For most compounds in the ketone form, the hydrogen bond energy is close to 12 kcal/mol. Form compounds in the enol form it is 8 kcal/mol. The exceptions are the compounds in the ketone form with COOH, CH_3_, Cl, and CH_2_CH_3_ substituents, in which the energies of the hydrogen bond are within the range of 8–19 kcal/mol.

### 2.10. Influence of the Substituent on the Transition State

To better understand the proton transfer between the ketone and enol form, the most probably transition state has been optimized. According to Rubén Marrero-Carballo et al. [15], proton transfer in the ketone form occurs from the methyl group in the middle ring to the oxygen in the carbonyl group to form the enol form. In the current work, the same proton pathway has been used but the effect of three different substituents in the middle ring on the transition state geometry and energy has additionally been investigated. Substituents that accept electrons from the middle ring or donate them to the ring have been chosen.

All substituents have subtle effects on the transition state energies (Table 6). The differences in the transition state energies of the hydrogen atom-substituted structure and the NO_2_ group-substituted structure are very similar. Despite the difference in the nature of the selected substituents, no direct interactions between the traveling proton and the substituents are observed.

For every transition state, the hydrogen bonds between the OH groups at the side rings and the CO group in the middle ring are broken. In the case of NH_2_ and OH substituents, the hydrogens from the side OH groups are directed toward each other (Figure 5). Only in the case of the NO_2_ substituent, both hydrogens from the OH groups face one direction. This is most likely due to the mesomeric effect of the NO_2_ group. In addition, the angle of the CO bond to the plane of the central ring constructed with four carbon atoms in common with the side aromatic rings has been investigated. The smallest angle is for the structure with the NO_2_ substituent, which pulls electrons out of the ring. On the other hand, the highest angle has been obtained when the substituent is the NH_2_ group, which is an electron donor.

### 2.11. Transition State for the Keto-Enol Reaction with Pyridine

Another proton transfer pathway has been carried out with the help of a pyridine molecule as a carrier [15] for the keto-enol reaction of 1,8-dihydroxy-9-anthrone substituted with NO_2_, OH, and H (Table 7). The lowest energy has been obtained for the initial state-pyridine with a substituted anthralin structure in the ketone form K + Pyr. In the first step, the proton from C10 is transferred to the nitrogen atom in the pyridine molecule. ΔG for the NO_2_ substituent is 17.60 kcal/mol and for the OH substituent ΔG is equal to 17.34 kcal/mol. The lowest transition state energy TS1 of 8.37 kcal/mol has been obtained for the hydrogen atom substituted structure. The next step on the reaction progress is the formation of the INT1 ion. The nitrogen in the pyridine moves toward the oxygen in the middle ring of the substituted anthralin to form a hydrogen bond. This results in the formation of an ion pair (INT2). In order for the protonated pyridine to release a proton into oxygen yielding the enol form E + Pyr, a low free energy is needed. In the case of the NO_2_ substituent, the ΔG is 13.93 kcal/mol, in the case of the OH group it is only 3.74 kcal/mol and for the H substituent-10.18 kcal/mol. For a better illustration of the reaction pathway involving pyridine, are provided figures showing each step of the reaction (Figure 6).

### 2.12. Theoretical Reaction Rate Constants

The rate constants in Table 8 illustrate how slow the conversion of the keto form to the enol form in 1,8-dihydroxy-9-anthrone molecules is. Comparison of the rate constants for the keto-enol reaction in substituted 1,8-dihydroxy-9-anthrones indicates that this reaction is not sensitive to substitution, except for the substitution of the NH_2_ group, which accelerates the investigated reaction. 

The addition of pyridine changes the reaction mechanism, making it a two-stage process, with the second stage clearly faster. In the case of both stages of the reaction, the effect of the substituent on the reaction rate is visible (Table 9), which is particularly evident in the case of the second stage of the reaction of the OH-substituted 1,8-dihydroxy-9-anthrone.

## 3. Conclusions

The lowest energy structure is the K6 tautomer, which represents the ketone form.

The intramolecular hydrogen bond affects the geometry of 1,8-dihydroxy-9-anthrone. The changes of the α angle, which is the angle between the planes of the side rings, and so representing the general shape of 1,8-dihydroxy-9-anthrone, is sensitive to the OHO hydrogen bonds.

The electronodonor and electronoacceptor properties of the substituent in the middle ring as well as the presence of intramolecular hydrogen bonds affects the antioxidant properties of 1,8-dihydroxy-9-anthrone. The greatest decrease in BDE(O-H) has been obtained for the structure with the NH_2_ substituent in the middle ring. Therefore, this compound can be expected to have the highest antioxidant properties.

The energy of the transition state in the keto-enol reaction in 1,8-dihydroxy-9-anthrone is unsensitive to the properties of the substituents in the central ring.

Taking into account the ΔH and the HOMA of the side aromatic rings, the preferred structure is the ketone form regardless the nature of the substituent in the middle ring of 1,8-dihydroxy-9-anthrone.

The keto-enol transformation in 1,8-dihydroxy-9-anthrone is significantly faster in the presence of pyridine.

## Data Availability

Xyz or wfn files are available on request from the corresponding author.

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
