# Peer review of "Prototropy, Intramolecular Interactions, Electron Delocalization, and Physicochemical Properties of 1,8-dihydroxy-9-anthrone—DFT-D3 Study of Substituent Effects"

_molecules, 2023, doi:10.3390/molecules28010344_

Round 1

Reviewer 1 Report

In this article, detailed and systematical studies have been carried out on the influencing factors of keto-enol equilibrium on dihydroxyanthrone derivative structures. The influencing factors of this system are discussed in depth, which has certain significance for the research of basic organic chemistry theory. However, for the novelty of the topic itself, the innovation of this paper is not strong. If the authors can further explore the complex molecular systems containing the dihydroxyanthrone structural unit, and compared more structures with existing crystal structures, it will be suitable to be published on molecules. 

Author Response

Structures from the CSD database are analyzed in the new paragraph 1.2, which has been added on page 5-6.

Reviewer 2 Report

please see the attached file

Reviewer 3 Report

This is an interesting manuscript describing the influence of the substituents on the keto-enol equilibrium of dihydroxyanthrone. Using DFT methods, the authors have calculated the O--H bond dissociation energies and obtained the angle (alpha) between the two side 6-membered rings for both keto and enol forms of dihydroxyanthrone with different substituents on C10. Based on the received data they described that the angle decreases with the increase in the number of hydrogen bonds in the molecule. Further, they mentioned that the angle in the enol form is smaller than in the ketone form and correlated the data with the HOMA values. Furthermore, they investigated the effect of the substituent on the BDE values of the O-H bonds at C9 and C-H bonds at C10 to shed light on their antioxidant properties. All in all, this contribution is a development of an established research direction (ref 1) and provides further development in the field. Therefore, I recommend the publication of this work in Molecules. Prior to publication, I recommend a careful review of the manuscript to refine the grammar, and spelling (singular form as well as plural form) be corrected throughout the manuscript. For example, dihydroxyanthron should be corrected as dihydroxyanthrone, preiously should be previously, and so on...

Minor points:-

Why compound 18 has a larger angle than 17, 37 than 36 and the enol form in 39 than the keto form 38. Is there any specific reason or they are just typos?

Author Response

The manuscript has been carefully reviewed and errors have been corrected

Reviewer 4 Report

Małgorzata and Irena presented a theoretical study to understand the influence of keto-enol equilibrium on the structure of dihy-droxyanthrone derivatives from the prespective of thermodynamic and kinetic. However, the author mainly focused on theomodynamical part, and the kinetic part is too weak, and the insightful analysis is not sufficient.

1. The decimal digits of energy should be unified.

2. The calculated reaction barrier of naked reaction is too high to be overcome, please add the calculation to use pyridine (or other base) as the proton transfer agent, otherwise, the discussion of the whole text is meaningless.

3. The temperature of thermodynamical parameters was lack, and the details of calculated method to obtain is also missed.

4. B3LYP-D3/6-311++G(d, p) is a primary method to estimate the thermodynamic of small system. Gerenally, the smaller system and the higher calculated level for quantum chemistry calculations. Therefore, the author need to confirm the accuracy of current method by the comparison with high level method, or add the single point energy calculation (using CCSD(T), DNLPO-CCSD(T), double-hybrid functional, and etc) based on optimized structures to obtain more accurate energy.

5. The wave function analysis by AIMALL is not sufficient to reveal the chemical nature of current system. I suggested the author to use multiwfn software to further analyze the electron donor/accepter effect of the substituents by electrostatic potentials, fukui function, atomic charge, and etc.

6. Normally, Nucleus Independent Chemical Shift (NICS) is the best method to estimate the aromaticity of a ring.

7. QTAIM method is usually a method to determine the preporty of a chemical bond or interaction by the parameter of BCP, which parameters are output by QTAIM, the parameters of ring critical point (RCP)? can you give the topology of structure including both BCP and RCP?

8. It's hard to distinguish C and H in Figure 3. Can the author use the ball model to illustrate the structures?

Author Response

Comments and Suggestions for Authors
Małgorzata and Irena presented a theoretical study to understand the influence of keto-enol equilibrium on the structure of dihy-droxyanthrone derivatives from the prespective of thermodynamic and kinetic. However, the author mainly focused on theomodynamical part, and the kinetic part is too weak, and the insightful analysis is not sufficient.

1. The decimal digits of energy should be unified.
Corrected

  1. The calculated reaction barrier of naked reaction is too high to be overcome, please add the calculation to use pyridine (or other base) as the proton transfer agent, otherwise, the discussion of the whole text is meaningless.

An additional paragraph 1.11 describing the keto-enol reaction involving pyridine has been added.

3. The temperature of thermodynamical parameters was lack, and the details of calculated method to obtain is also missed.

Description of the calculation method has been adequately supplemented.

4. B3LYP-D3/6-311++G(d, p) is a primary method to estimate the thermodynamic of small system. Gerenally, the smaller system and the higher calculated level for quantum chemistry calculations. Therefore, the author need to confirm the accuracy of current method by the comparison with high level method, or add the single point energy calculation (using CCSD(T), DNLPO-CCSD(T), double-hybrid functional, and etc) based on optimized structures to obtain more accurate energy.

We remember about the influence of the calculation method used on the optimization result and the energy value. However, our goal was to compare our results with those obtained previously [ref. 14] and therefore we decided to use the same method.

  1. The wave function analysis by AIMALL is not sufficient to reveal the chemical nature of current system. I suggested the author to use multiwfn software to further analyze the electron donor/accepter effect of the substituents by electrostatic potentials, fukui function, atomic charge, and etc.

In Section 1.6 “Dipole Moment, Average Local Ionization Energy, Electrostatic Potential" we analyze additional structural parameters obtained using the multiwfn program.

6. Normally, Nucleus Independent Chemical Shift (NICS) is the best method to estimate the aromaticity of a ring.

NICS is very commonly used parameter to describe aromaticity, but the rings analyzed in this work are very far from planarity so NICS could give false information

7. QTAIM method is usually a method to determine the preporty of a chemical bond or interaction by the parameter of BCP, which parameters are output by QTAIM, the parameters of ring critical point (RCP)? can you give the topology of structure including both BCP and RCP?

The parameter values in the RCP are not sensitive enough to describe changes in electron density in the rings.

8. It's hard to distinguish C and H in Figure 3. Can the author use the ball model to illustrate the structures?

The Figure has been corrected.

Round 2

Reviewer 1 Report

Compared with the first edition, the revised manuscript has been much improved. I believe the current revised manuscript can be considered for acceptance.

Author Response

Corrected

Reviewer 4 Report

The author has adressed my concernings, but there are some minior problem before acceptance. 

1. The unit of energy in many tabled is not provided

2. one can use kcal/mol or kcal mol-1, but you cannot use both of them in one paper

3. the superscript of kcal mol-1 was ignored

4. Could you please move some less improtant figures or tables into Supporting Information?

Author Response

Comments and Suggestions for Authors

The author has adressed my concernings, but there are some minior problem before acceptance. 

1. The unit of energy in many tabled is not provided

Corrected

2. one can use kcal/mol or kcal mol-1, but you cannot use both of them in one paper

Corrected

3. the superscript of kcal mol-1 was ignored
Corrected

  1. Could you please move some less improtant figures or tables into Supporting Information?

We think this is not a good idea. The text reads badly when you have to refer to the supplement in many situations.

Round 3

Author Response

We have made another attempt to optimize the K3 tautomer using DFT-D3 B3LYP/6-311++G** level. Despite blocking the bonds, the K3 tautomer optimized to the K6 tautomer as we described in the manuscript. All parameters in Table 2 and in the text have been checked again.